# Too fast to stay on track? Shorter time to first anti-retroviral regimen is not associated with better retention in care in the French Dat'AIDS cohort

L. Cuzin[1,2]*, L. Cotte[3], C. Delpierre[2], C. Allavena[4], M-A. Valantin[5], D. Rey[6], P. Delobel[7,8], P. Pugliese[9], F. Raffi[4,10], A. Cabié[1,11], on behalf of the Dat'AIDS Study group¶

1 Infectious and Tropical Diseases Unit, University Hospital of Martinique, Fort de France, France, 2 INSERM UMR1017, Toulouse III University, Toulouse, France, 3 Infectious Disease Unit, Hospices Civils de Lyon, Lyon, France, 4 Infectious Diseases Unit, Hotel Dieu University Hospital of Nantes, Nantes, France, 5 Infectious Diseases Unit, University Hospital of Pitié-Salpêtrière, Paris, France, 6 HIV Infection Care Centre, University Hospital, Strasbourg, France, 7 Infectious and Tropical Diseases Unit, Purpan University Hospital, Toulouse, France, 8 INSERM U1043 - CNRS UMR 5282, Centre de Physiopathologie Toulouse-Purpan, Toulouse, France, 9 Infectious Diseases Unit, University Hospital of Nice, Nice, France, 10 CIC 1413, INSERM, Nantes, France, 11 CIC1424, INSERM, Fort-de-France, France

¶ Membership of the Dat'AIDS study group is provided in the Acknowledgments.
* lise.cuzin@chu-martinique.fr

**Data Availability Statement:** All relevant data are within the paper and its Supporting Information files.

## Abstract

### Background

Rapid antiretroviral therapy (ART) initiation has been proven beneficial for patients and the community. We aimed to analyze recent changes in timing of ART initiation in France and consequences of early start.

### Methods

We selected from a prospective nationwide cohort, on 12/31/2017, patients with HIV-1 infection diagnosed between 01/01/2010 and 12/31/2015. We described time from (1) diagnosis to first specialized medical encounter, (2) from this encounter to ART initiation, (3) from diagnosis to first undetectable HIV viral load (VL). We analyzed the determinants of measured temporal trends. A multivariate logistic regression was performed to assess characteristics related with 1-year retention in care.

### Results

In the 7 245 included patients, median time (1) from HIV diagnosis to first medical encounter was 13 (IQR: 6–32) days, (2) to ART initiation was 27 (IQR: 9–91) days, decreasing from 42 (IQR: 13–272) days in 2010 to 18 (IQR: 7–42) in 2015 (p<0.0001), (3) to first undetectable VL was 257 (IQR: 151–496) days, decreasing from 378 (IQR: 201–810) days in 2010 to 169 (IQR: 97–281) in 2015. After one year, proportion of patients alive and still in care was significantly lower in those in the lower quartile of time from first encounter to ART (<9 days) than those in the higher quartile (>90 days), 79.9% and 85.2%, respectively (p<0.0001).

**Funding:** The authors received no specific funding for this work.

**Competing interests:** The authors have declared that no competing interest exist.

## Conclusions

In a country with unrestricted rapid access to ART, keeping recently diagnosed HIV infected patients in care remains challenging. Starting ART rapidly did not seem to be profitable for all and every patient.

## Introduction

Early initiation of anti-retroviral therapy (ART) for people living with human immunodeficiency virus (PLWH) reduces individual morbidity and mortality [1–3]. It also leads to a shortened period of possible HIV transmission by suppressing viral replication [4]. Thus, national and international guidelines recommend ART for all PLWH without any restrictions of CD4 cell counts (https://www.who.int/hiv/topics/treatment/en/). Most recent guidelines recommend starting ART as soon as possible after diagnosis, including immediately after diagnosis, unless the patient is not ready to commit to starting therapy, or in specific cases such as cryptococcosis or tuberculosis meningitidis—because of possible immune reconstitution inflammatory syndrome [5]. Same day ART initiation after HIV disclosure has been proven to be feasible,[4] even in very deprived settings [6]. However, structural barriers, patients' unmet basic needs [7] and patients' and providers' attitudes may impede the rapid initiation of ART [4].

The French national health system allows every PLWH to access ART free of charge with no administrative delay. Nevertheless, it has been described that the timing of initiation was different between geographical regions [8].

Our aim was, using a nationwide prospectively collected database, to describe determinants of the time from the first encounter with an HIV clinician after HIV diagnosis and the first ART prescription, changes in the recent years and the influence of an early ART start on 1-year retention in care.

## Patients and methods

Information was collected from 23 centers in France (including overseas territories) participating in the Dat'AIDS cohort (Clinicaltrials.gov ref. NCT02898987). The authors did not have access to any patient identifying information as part of this work. For the purpose of this study, we selected patients who were included in the cohort with a newly diagnosed HIV-1 infection between January the 1st, 2010 and December 31, 2015. We collected the date of HIV diagnosis, the date of first medical encounter with an HIV clinician (thereafter "first encounter"), the date of first ART prescription, the choice of 3rd ART drug (boosted protease inhibitor-bPI, non nucleosidic reverse transcriptase inhibitor-NNRTI, integrase inhibitor-INSTI, or other regimen), the date of last news, CD4 cells counts and plasma HIV viral load (VL) at the time of diagnosis and at the time of ART initiation, age at diagnosis, sex and most probable route of HIV acquisition. During the patients' follow-up, we collected VL at six and 12 months after ART initiation, date of first non-detectable VL (<50 copies/mL), date of and reason for initial ART modification, and vital status. French recommendations include a clinical visit one and three months after ART initiation, and twice a year thereafter. One-year retention in care was defined as "yes" if at least one visit was recorded between 9 and 15 months after ART initiation, otherwise as "no". Because of a strong interaction between sex and the most probable

route of HIV acquisition, the patients were classified in four groups for the analyses: men having sex with men (MSM), men having sex with women (MSW), transgender and women.

Only patients with reliable dates of first encounter, first ART prescription and first non-detectable VL were included in the analysis. Two time lapses were first calculated: time from HIV diagnosis to first encounter with an HIV clinician and time from this first encounter to first ART prescription. Additionally, time from diagnosis to first non-detectable VL was also calculated, with the aim of approaching individual and collective benefits of an early ART.

Because the time from first encounter to first ART varied in the recent years and may also vary depending on patients' characteristics, we first described the patients following the year of diagnosis and following CD4 cell counts at diagnosis. The calculated time lapses were also described following patients' CD4 cell count at diagnosis and year of HIV diagnosis. Patients' evolution while receiving ART was described following the time from first encounter to first ART prescription, in classes following quartiles. Finally, we performed a multivariate logistic regression looking for characteristics related with 1-year retention in care, in which we included all characteristics significantly (p<0.10) related with 1-year retention in care in the univariate analysis. Data were extracted on December 31, 2017 as to ensure a minimum follow-up, even in patients with long delay to ART initiation.

Quantitative variables are described by medians and $1^{st}$-$3^{rd}$ quartiles (IQR), and compared by Kruskal-Wallis tests. Qualitative data are described by frequencies and compared by $Chi^2$ tests. Statistical analyses were performed using R (R.app GUI 1.70, S. Urbanek & H.-J. Bibiko, R Foundation for Statistical Computing, 2016).

## Results

From 9 680 patients fulfilling the selection criteria, 2 435 were excluded from the analysis because the dates of interest were not reliable. Patients characteristics for both populations are shown in Table 1.

In the analyzed population, the median time from HIV diagnosis to first encounter was 13 (IQR 6–32) days. Overall the median time from first encounter to first ART prescription was 27 (IQR 9–91) days, decreasing from 42 (IQR 13–272) days in 2010 to 18 (IQR 7–42) days in 2015 (p<0.0001) as shown in Table 2. Meanwhile patients' characteristics also varied across

**Table 1. Patients characteristics, with comparison of the included and excluded populations.**

| | | Included N = 7245 | Excluded N = 2435 | P |
|---|---|---|---|---|
| Sex (M/W) | | 5300 (72%) / 1945 (28%) | 1672 (68.7%) / 763 (31.3%) | <0.0001 |
| Route of acquisition | Men having sex with men | 3388 (46.8%) | 981 (40.3%) | <0.0001 |
| | Men having sex with women | 3260 (45.0%) | 1132 (46.5%) | |
| | IVDU[a] | 90 (1.2%) | 46 (1.9%) | |
| | Other/unknown | 507 (7%) | 276 (11.3%) | |
| Diagnosis during Acute HIV infection | | 788 (10.8%) | 188 (7.7%) | <0.0001 |
| Age at HIV diagnosis (years, median IQR) | | 36 (28–46) | 34 (28–43) | <0.0001 |
| Age at ART prescription (years, median IQR) | | 40 (32–50) | 39 (32–48) | 0.002 |
| CD4 cell count at HIV diagnosis (/μL, median IQR) | | 362 (198–537) | 295 (121–475) | <0.0001 |
| HIV viral load at HIV diagnosis (log.copies/mL, median IQR) | | 4.74 (4.1–5.3) | 4.9 (4.2–5.5) | <0.0001 |
| CD4 cell count at ART prescription (/μL, median IQR) | | 352 (197–505) | 297 (124–470) | <0.0001 |
| HIV viral load at ART prescription (log.copies/mL, median, IQR) | | 4.7 (4.1–5.2) | 4.9 (4.3–5.5) | <0.0001 |

[a]: Intravenous drug use.

**Table 2. Patients characteristics across the study period and studied time lapses, by year of HIV diagnosis.**

| Year of HIV diagnosis | | 2010 N = 1192 | 2011 N = 1204 | 2012 N = 1196 | 2013 N = 1173 | 2014 N = 1280 | 2015 N = 1200 | P |
|---|---|---|---|---|---|---|---|---|
| Age at diagnostic (years; median, IQR) | | 36 (29–44) | 37 (29–45) | 36 (28–47) | 36 (28–47) | 36 (28–46) | 36 (28–47) | 0.95 |
| Women (%) | | 28.4 | 26.2 | 27.0 | 26.9 | 26.6 | 26 | 0.11 |
| Men having sex with Men (%) | | 42.9 | 46.6 | 45.2 | 47.5 | 44.8 | 48.8 | |
| Men having sex with Women (%) | | 27.4 | 26.8 | 26.7 | 25.1 | 27.6 | 24.7 | |
| Transgender Men to Women (%) | | 1.3 | 0.3 | 1.1 | 0.5 | 1.0 | 0.6 | |
| CD4 Cell count/ μL at diagnosis (median, IQR) | | 345 (186–517) | 356 (190–541) | 374 (221–540) | 369 (202–558) | 373 (203–539) | 361 (187–538) | 0.11 |
| CD4 Cell count/ μL at first ART initiation (median, IQR) | | 327 (188–456) | 343 (188–482) | 356 (208–507) | 362 (198–532) | 372 (208–545) | 356 (190–529) | <0.0001 |
| ART 3rd drug | bPI[a] (%) | 58.8 | 63.4 | 59.5 | 59.7 | 47.0 | 41.2 | <0.0001 |
| | NNRTI[b] (%) | 26.9 | 23.2 | 24.3 | 23.3 | 21.4 | 13.8 | |
| | INSTI[c] (%) | 7.8 | 7.2 | 8.5 | 10.4 | 26.0 | 40.7 | |
| | Other (%) | 6.5 | 6.2 | 7.7 | 6.6 | 5.6 | 4.3 | |
| Alive and in care after 12 months of ART (%) | | 83.4 | 83.2 | 83.9 | 83.7 | 83.8 | 85.1 | 0.86 |
| From diagnosis to first visit (days, median IQR) | | 17 (7–55) | 15 (7–42) | 14 (6–36) | 13 (6–25) | 11 (5–26) | 10 (4–24) | <0.0001 |
| From first visit to ART (days, median IQR) | | 42 (13–272) | 41 (14–212) | 35 (12–147) | 25 (8–72) | 21 (6–49) | 18 (7–42) | <0.0001 |
| From diagnosis to undetectable VL (days, median IQR) | | 378 (201–810) | 339 (195–734) | 300 (190–600) | 257 (167–436) | 198 (125–330) | 169 (97–281) | <0.0001 |

[a]: boosted protease inhibitor

[b]: non-nucleosidic reverse transcriptase inhibitor

[c]: integrase inhibitor.

the study period. CD4 cell counts at the time of first ART prescription increased from 327 (IQR 188–456) cells/ μL for PLWH diagnosed in 2010 to 356 (IQR 190–529) cells/ μL for those diagnosed in 2015 (p<0.0001).

The median time from first encounter to first ART prescription also varied according to CD4 cell count at the time of diagnosis, from a median of 14 days (IQR 7–27) in patients with less than 200 cells/ μL to 80 days (IQR 18–363) in those with more than 500 cells/ μL (see Table 3).

Overall the median time from HIV diagnosis to first VL below 50 copies/mL was 257 (IQR 151–496) days, decreasing from 378 (IQR 201–810) in for PLWH diagnosed in 2010 to 169 days (IQR 97–281) for those diagnosed in 2015 (p<0.0001). Overall, VL was non-detectable for 74.8% and 81.1% of the patients after 6 and 12 months of ART, respectively. Proportion of patients with a non-detectable VL after 6 months of ART increased from 66.8% for patients diagnosed in 2010 to 83.6% for those diagnosed in 2015 (p<0.0001). Similarly, proportion of patients with a non-detectable VL after 12 months of ART increased from 75.4% for patients diagnosed in 2010 to 83.4% for those diagnosed in 2015 (p = 0.04). At month 12, 83.8% of the patients were actively in care, and 2.1% were dead.

The first ART was modified, after a median time of 18 (IQR 6–37) months, in 6 985 (96.4%) patients. Reason for modification was simplification (2 036 patients, 28.1%), adverse event (1 347, 18.6%), virological failure (198, 2.7%), patients' death (147, 2.1%) or other/unknown

**Table 3. Patients characteristics and studied time lapses, by CD4 cell count at the time of HIV diagnosis.**

| CD4 cell count at HIV diagnosis/ µL | | <200 N = 1594 | 200–350 N = 1589 | 350–500 N = 1593 | >500 N = 1588 | P |
|---|---|---|---|---|---|---|
| Age at diagnosis (Years, median, IQR) | | 41 (33–51) | 37 (29–47) | 34 (27–44) | 34 (27–43) | <0.0001 |
| End of study (%) | In care | 75.2 | 76.4 | 75.5 | 77.8 | <0.0001 |
| | Changed place of care | 7.7 | 9.2 | 10.7 | 9.3 | |
| | Lost to follow-up | 11.8 | 12.6 | 12.9 | 12.3 | |
| | Dead | 5.3 | 1.8 | 0.9 | 0.6 | |
| Sex and way of acquisition (%) | MSM[a] | 15.8 | 23.8 | 29.1 | 31.3 | <0.0001 |
| | MSW[b] | 37.7 | 25.6 | 20.1 | 16.7 | |
| | Women | 29.6 | 26.4 | 22.4 | 21.6 | |
| | Trans gender M>W | 19.5 | 29.3 | 31.7 | 19.5 | |
| ART 3rd drug (%) | bPI[c] | 69.7 | 58.0 | 51.2 | 44.2 | <0.0001 |
| | NNRTI[d] | 8.5 | 22.0 | 25.7 | 30.7 | |
| | INSTI[e] | 13.7 | 14.5 | 17.2 | 20.1 | |
| | Other | 8.1 | 5.5 | 5.9 | 5.0 | |
| From diagnosis to first visit (days, median IQR) | | 9 (3–19) | 12 (6–22) | 12 (5–22) | 13 (6–27) | <0.0001 |
| Time from first visit to ART (days, median, IQR) | | 14 (7–27) | 21 (7–56) | 42 (14–144) | 80 (18–364) | <0.0001 |
| From diagnosis to undetectable VL (days, median IQR) | | 228 (150–300) | 212 (132–336) | 239 (140–419) | 289 (142–634) | <0.0001 |

[a]: Men having sex with men

[b]: men having sex with women

[c]: boosted protease inhibitor

[d]: non-nucleosidic reverse transcriptase inhibitor

[e]: integrase inhibitor.

reasons (1 540, 21.2%). Among patients whose regimen was simplified, 92.4% were retained in care at month 12, as compared to 89.4% of the patients whose regimen was modified due to adverse events (p<0.0001).

Evolution of the patients after ART initiation, following the quartiles of time from first encounter to first ART prescription, is described in Table 4. One-year retention in care was better for patients with a longer period between first medical encounter and ART initiation (from 79.9% in the first quartile to 85.2% in the 4th quartile, p<0.0001). One-year retention in care did not differ according to the choice of third agent: 83.5% with INNTIs, 81.8% with INSTIs, 84.4% wit bPI, 85.8% for other regimens, p = 0.11. At the end of the study period the

**Table 4. Patients' evolution depending on the time from first medical visit to first ART prescription (classes by quartiles).**

| Time from first medical visit to first ART (days) | | < 9 N = 1881 | 9–27 N = 1784 | 28–90 N = 1800 | > 90 N = 1780 | P |
|---|---|---|---|---|---|---|
| Alive and in care at month 12 after ART prescription (%) | | 79.9 | 84.5 | 85.9 | 85.2 | <0.0001 |
| Time from diagnosis to undetectable VL (days; median, IQR) | | 194 (108–351) | 210 (130–361) | 232 (152–357) | 527 (311–924) | <0.0001 |
| Length of first ART (months; median, IQR) | | 14 (5–32) | 17 (7–35) | 21.5 (7–39) | 22 (7–42) | <0.0001 |
| End of study situation | Dead (%) | 2.2 | 3.1 | 1.9 | 0.9 | 0.002 |
| | LTFU (%) | 14.3 | 12.4 | 13.4 | 12.5 | 0.002 |
| VL < 50 copies/mL after 6 months of ART (%) | | 72.1 | 69.8 | 78.9 | 79.6 | <0.0001 |
| VL < 50 copies/mL after 12 months of ART (%) | | 78.0 | 81.5 | 84.1 | 81.4 | 0.12 |
| VL < 50 copies/mL after 18 months of ART (%) | | 83.7 | 85.5 | 86.9 | 87.9 | 0.15 |

**Table 5. Baseline characteristics related with one-year retention in care after starting ART (uni and multivariate logistic regression, HR: hazard ratio, aHR: adjusted hazard ratio, CI95%: 95% confidence interval).**

| | | HR | CI95% | P | aHR | CI95% | P |
|---|---|---|---|---|---|---|---|
| Age at HIV diagnosis* | < 28 y | Reference | | | Reference | | |
| | 28–36 y | 1.39 | 1.18–1.65 | 0.0001 | 1.44 | 1.21–1.71 | <0.0001 |
| | 37–46 y | 1.56 | 1.32–1.86 | <0.0001 | 1.67 | 1.40–2.00 | <0.0001 |
| | > 46 y | 1.59 | 1.33–1.89 | <0.0001 | 1.78 | 1.48–2.14 | <0.0001 |
| Sex/route of acquisition | Women | Reference | | | Reference | | |
| | MSW[a] | 0.79 | 0.67–0.93 | 0.007 | 0.72 | 0.61–0.86 | 0.0001 |
| | MSM[b] | 1.09 | 0.93–1.28 | 0.3 | 1.10 | 0.94–1.28 | 0.94 |
| | Trans M>W[c] | 0.60 | 0.33–1.15 | 0.10 | 0.62 | 0.34–1.19 | 0.32 |
| CD4 cell count at diagnosis* | <200 | Reference | | | | | |
| | 200–350 | 1.22 | 1.00–1.50 | 0.04 | | | |
| | 351–500 | 1.13 | 0.92–1.38 | 0.22 | | | |
| | >500 | 0.98 | 0.80–1.19 | 0.85 | | | |
| Year of HIV Diagnosis | 2010 | Reference | | | | | |
| | 2011 | 0.98 | 0.79–1.22 | 0.9 | | | |
| | 2012 | 1.03 | 0.83–1.29 | 0.7 | | | |
| | 2013 | 1.02 | 0.82–1.27 | 0.8 | | | |
| | 2014 | 1.03 | 0.83–1.27 | 0.8 | | | |
| | 2015 | 1.14 | 0.91–1.42 | 0.2 | | | |
| Time from first visit to ART (days)* | < 9 days | Reference | | | Reference | | |
| | 9–27 | 1.37 | 1.16–1.63 | <0.0001 | 1.36 | 1.14–1.61 | <0.0001 |
| | 28–90 | 1.53 | 1.29–1.82 | <0.0001 | 1.52 | 1.28–1.82 | <0.0001 |
| | > 90 | 1.45 | 1.22–1.72 | <0.0001 | 1.48 | 1.24–1.76 | <0.0001 |

*: classes following quartiles

a: Men having sex with women

b: men having sex with men

c: transgender men to women

proportion of patients lost-to-follow-up was higher in patients with rapid ART initiation (from 14.3% in the first quartile to 12.5% in the 4th quartile, p<0.0001). The proportion of patients with an undetectable VL 6 months after ART initiation was lower in case of rapid ART initiation (from 72.1% in the first quartile to 79.6% in the 4th quartile, p<0.0001).

Uni- and multi-variate logistic regressions (see Table 5) showed that higher age at HIV diagnosis and a longer time between first encounter and first ART prescription were associated with a better 1-year retention. On the opposite, MSW were less likely to be retained in care than women. Notably CD4 cell count at diagnosis was not related with retention in care at month 12.

## Discussion

In this large nationwide prospectively followed population, we observed that the median time between HIV diagnosis and first encounter with an HIV clinician significantly decreased from a median of 17 days for patients diagnosed in 2010 to 10 days for those diagnosed in 2015. Time between this first encounter and first ART prescription also decreased from a median of 42 to 18 days. This reflects changes in national guidelines that since 2013 have been recommending treatment for all PLWH regardless of CD4 cell counts [9]. Besides historic evolution,

the time from first medical encounter to ART prescription also depended on the severity of the disease at the time of diagnosis. Starting early has been described to be related with a decrease in mortality [10], a better CD4 cell count recovery [11] and a better retention in care [12, 13]. In these studies, an early start was defined as 1 month [10], 6 months [11] or one day when feasible [12]. Notably in our population diagnosed in 2015, 25% received their first ART prescription before 7 days after first medical encounter, 50% within 18 days and 75% before 42 days.

Rapid ART initiation also has the potential to bring down the period of possible HIV transmission, and is part of what is needed to reach the goal of curbing the epidemic [14]. Furthermore, French national recommendations have been recommending to use an INSTI-based first regimen since 2014. In addition to more widely use of INSTI in first-line ART, early ART initiation led to shortening by more than half the time between HIV diagnosis and first non-detectable viral load in our population across the study period. This is an important public health finding on a community level, but has to be balanced with the proportion of patients not pursuing care and treatment.

In our population, initiation of a first anti-retroviral treatment within 9 days was negatively associated both with retention in care 12 months later and with being still in care at the end of the study period. These patients not retained in care will necessarily come back to care, and such care interruptions have been described as detrimental [15]. Other characteristics related with retention in care have already been described by others, such as age and men having sex with men [16–18]. The poorer retention rate in men having sex with women could be related to some specific factors, more prevalent in this population, such as migration, stigmatization, deny, and/or socio-psychological situation. It is possible that some characteristics not taken in account in our analyses played a significant role, specifically the social and life context of populations, so this question needs further investigation. Absence of improvement in retention in care with early ART start has recently been described with the RAPID initiative in San Francisco [4, 14], where retention was not better in patients starting ART on the day of HIV diagnosis. In the Chinese national program, among patients starting ART within 30 days after HIV diagnosis, starting earlier than 30 days was not related with a better retention in care [13]. The French health and social system allows all HIV infected patients to access care free of charge without any delay, so 75% of the studied patients had been starting ART within 90 days. In contrast, in the START study, an early start was defined as starting ART in the first 4 months [19]. We can hypothesize that as our entire cohort has been receiving ART in a relatively short time, going faster than already fast did not bring any advantage regarding retention in care. Indeed, if patients are not prepared, educated and motivated on HIV treatment, advantages and constraints, starting ART too early could be counterproductive, stressing the need for qualitative studies on the reasons for remaining in care. It is possible that some populations, as MSM or women, have more prior personal interest in being treated early because they are aware of the individual and collective benefits of having an undetectable VL. On the contrary, MSW may be less informed before the announcement of their infection, thus less motivated, and in need of a longer time to understand the importance of care and treatment. Thus, they may drop off due to lack of information or interest. In addition, barriers to an early start are present in some patients, in particular the need to exclude latent or active tuberculosis which is prevalent in our cohort that includes patients both from Sub Saharian Africa and the Caribbean Islands [20]. Since ART is initiated for life and since the process of preparing patients to accept such a treatment and address barriers to treatment can vary notably, we consider, based on the results of our study, that an early start of ART should not be a hasty start of ART. Waiting 2 to 4 weeks after first HIV clinician encounter to initiate ART was associated with a significantly better 1-year retention.

We observed that while the median duration of first ART regimen was of 18 months, 25% of the patients changed their regimen during the first 6 months. The major reason for changing the regimen was treatment simplification, mainly due to the availability of new single tablet regimens. ART simplification was associated with a better retention in care, while changing ART because of intolerance was not. This observation could be explained both ways. Either physicians chose to propose simple treatments to well-adherent patients, or on the other end patients suffering from adverse events may have a negative perception and be discouraged to continue care. Anyhow, the relation between ART modifications and retention in care needs further evaluation.

The strength of our study is the large prospectively followed population. Nevertheless, we have limitations. First some important patients' characteristics that may be related with retention in care could not be taken in account in this analysis. Second, it has been reported that in some French areas, natives from sub Saharan Africa had delayed access to care [20]. We could not analyze place of birth because of missing information in more than 25% of the population. Third, the socio-economic status of the patients constituting the cohort is known to be very different between the participating centers [21], but we could not find any association between the geographical region of the care center and one-year retention in care else the already known difference between mainland and overseas territories (See S1 Table). The relationship between social deprivation and retention in care in France remains to be studied.

In conclusion, in a country with unrestricted rapid access to ART, the time from first medical encounter to first ART significantly decreased in the recent years. Evolving recommendations and INSTI-based initial regimens significantly reduced the time from HIV diagnosis to first undetectable viral load. Nonetheless, keeping recently diagnosed HIV infected patients in care still remains challenging. Starting treatment too rapidly does not seem to be profitable for all and every patient.

## Supporting information

**S1 Dataset.**
(XLSX)

**S1 Table. Proportion of patients in care at month 12 by region of care.**
(DOCX)

## Acknowledgments

Lise Cuzin designed the research, did the analysis and wrote the first draft, Laurent Cotte, Clotilde Allavena, Marc-Antoine Valantin, David Rey, Pierre Delobel, Pascal Pugliese were responsible for the data quality in the centers, provided advice along the study and participated in the manuscript edition, Cyrille Delpierre provided advice on the analysis and the final presentation of results and discussion, François Raffi gave important input on the analysis and the redaction, Andre Cabié was responsible for the data in his center and gave useful advice during the paper's redaction.

We are indebted to the technical staff of each site for controlling the quality of the data, especially to Thomas Jovelin for data extraction (Nantes, France).

The prospective database is collected via the Nadis® electronic medical record (Fedialis Medica, Advanced Biological Laboratories, France).

* **Dat'AIDS Study Group**: **Lead** Pr A. Cabié, Martinique university Hospital, Fort de France, FWI. Andre.cabie@chu-martinique.fr. Members: C. Drobacheff-Thiébaut, A. Foltzer, K. Bouiller, L. Hustache- Mathieu, C. Chirouze, Q. Lepiller, F. Bozon, O Babre, A.S. Brunel, P.

Muret (Besançon University Hospital); H. Laurichesse, O. Lesens, M. Vidal, N. Mrozek, C. Aumeran, O. Baud, V. Corbin, P. Letertre-Gibert, S. Casanova, J. Prouteau, C. Jacomet (Clermont Ferrand University Hospital); I. Lamaury, I. Fabre, E. Curlier, R. Ouissa, C. Herrmann-Storck,B. Tressieres, T. Bonijoly, M.C. Receveur, F. Boulard, C.Daniel, C. Clavel (Guadeloupe University Hospital); D. Merrien, P. Perré, T. Guimard, O. Bollangier, S. Leautez, M. Morrier, L. Laine (La Roche sur Yon Hospital); F. Ader, A. Becker, F. Biron, A. Boibieux, L. Cotte, T. Ferry, P Miailhes, T. Perpoint, S. Roux, C. Triffault-Fillit, S. Degroodt, C. Brochier, F Valour, C. Chidiac (Lyon University Hospital); A. Ménard, A.Y. Belkhir, P.Colson, C. Dhiver, A. Madrid, M. Martin-Degiovani, L. Meddeb, M. Mokhtari, A.Motte, A.Raoux, I. Ravaux, C. Tamalet, C. Toméi, H. Tissot Dupont (Marseille IHU Méditerranée); S. Brégigeon, O. Zaegel-Faucher, V. Obry-Roguet, H. Laroche, M. Orticoni, M.J. Soavi, P. Geneau de Lamarlière, E. Ressiot, M.J. Ducassou, I. Jaquet, S. Galie, A. Galinier, P. Martinet, M. Landon, A.S. Ritleng, A. Ivanova, C. Debreux, C. Lions, I. Poizot-Martin (Marseille Ste Marguerite Hospital); O. Cabras, L. Cuzin, K. Guitteaud, M. Illiaquer, S. Pierre-François, L. Osei, J. Pasquier, K. Rome, E. Sidani, J.M. Turmel, C. Varache, A. Cabié (Martinique University Hospital); N. Atoui, M. Bistoquet, E Delaporte, V. Le Moing, A. Makinson, N. Meftah, C. Merle de Boever, B. Montes, A. Montoya Ferrer, E. Tuaillon, J. Reynes (Montpellier University Hospital); M. André, L. Boyer, MP. Bouillon, M. Delestan, C. Rabaud, T. May, B. Hoen (Nancy University Hospital); C. Allavena, C. Bernaud, E. Billaud, C. Biron, B. Bonnet, S. Bouchez, D. Boutoille, C. Brunet-Cartier, C. Deschanvres, B. Gaborti, N. Hall, T. Jovelin, P. Morineau, V. Reliquet, S. Sécher, M. Cavellec, A. Soria, V. Ferré, E. André-Garnier, A. Rodallec, M. Lefebvre, O. Grossi, O. Aubry, F. Raffi (Nantes University Hospital); P. Pugliese, S. Breaud, C. Ceppi, D. Chirio, E. Cua, P. Dellamonica, E. Demonchy, A. De Monte, J. Durant, C. Etienne, S. Ferrando, R. Garraffo, C. Michelangeli, V. Mondain, A. Naqvi, N. Oran, I. Perbost, S. Pillet, C. Pradier, B. Prouvost-Keller, K. Risso, V. Rio, PM. Roger, E. Rosenthal, S. Sausse, I. Touitou, S. Wehrlen-Pugliese, G. Zouzou (Nice University Hospital); L. Hocqueloux, T. Prazuck, C. Gubavu, A. Sève, A. Maka, C. Boulard, G. Thomas (Orleans University Hospital); E. Botelho-Nevers, A. Gagneux-Brunon, A. Frésard, V. Ronat, F. Lucht (St Etienne); P. Fischer, M. Partisani, C. Cheneau, M Priester, ML Batard, C Bernard-Henry, E. de Mautort, S. Fafi-Kremer, D. Rey (Strasbourg); M. Alvarez, N. Biezunski, A. Debard, C. Delpierre, P. Lansalot, L. Lelièvre, G. Martin-Blondel, M. Piffaut, L. Porte, K. Saune, P. Delobel (Toulouse University Hospital); F. Ajana, E. Aïssi, I. Alcaraz, V. Baclet, L. Bocket, A. Boucher, P. Choisy, T. Huleux, B. Lafon-Desmurs, A. Meybeck, M. Pradier, O. Robineau, N. Viget, M. Valette (Tourcoing); Y. Yazdanpanah, R. Landman, C. Duvivier, M.A. Valantin, R. Agher, C. Katlama, O. Lortholary, V. Avettand-Fenoel, C. Rouzioux, P.H. Consigny, G. Cessot, F. Touam, R. Usubillaga, K. Benhadj (Paris University Hospitals).

## Author Contributions

**Conceptualization:** L. Cuzin, P. Delobel, F. Raffi, A. Cabié.

**Data curation:** L. Cuzin, P. Pugliese.

**Formal analysis:** L. Cuzin, C. Delpierre.

**Investigation:** L. Cotte, D. Rey, P. Delobel, A. Cabié.

**Methodology:** C. Delpierre, M-A. Valantin, A. Cabié.

**Software:** P. Pugliese.

**Supervision:** L. Cotte, C. Allavena, M-A. Valantin, D. Rey, F. Raffi, A. Cabié.

**Validation:** L. Cuzin, C. Delpierre, C. Allavena, D. Rey, P. Delobel, F. Raffi, A. Cabié.

**Writing – original draft:** L. Cuzin.

**Writing – review & editing:** L. Cuzin, L. Cotte, C. Delpierre, C. Allavena, M-A. Valantin, D. Rey, P. Delobel, P. Pugliese, F. Raffi, A. Cabié.

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
