## [Decision Letter · Decision Letter 0]

14 Aug 2019

PONE-D-19-18078

Too fast to stay on track? Shorter time to first anti-retroviral regimen is not associated with better retention in care in the French Dat’AIDS cohort

PLOS ONE

Dear Dr. Cuzin,

Thank you for submitting your manuscript to PLOS ONE. After careful consideration, by one reviewer and myself, we feel that it has merit but does not fully meet PLOS ONE’s publication criteria as it currently stands. Therefore, we invite you to submit a revised version of the manuscript that addresses the points raised during the review process.

Specifically, both myself and the reviewer felt that there was only minimal discussion on the possible reasons for early ART initiation relating to lower retention rates. This may be discussed further with additional speculations.

We would appreciate receiving your revised manuscript by Sep 28 2019 11:59PM. To enhance the reproducibility of your results, we recommend that if applicable you deposit your laboratory protocols in protocols.io, where a protocol can be assigned its own identifier (DOI) such that it can be cited independently in the future. For instructions see: http://journals.plos.org/plosone/s/submission-guidelines#loc-laboratory-protocols

We look forward to receiving your revised manuscript.

Kind regards,

Cristian Apetrei, MD, PhD

Academic Editor

PLOS ONE

Journal Requirements:

3. One of the noted authors is a group or consortium: Dat’AIDS Study group. In addition to naming the author group, please list the individual authors and affiliations within this group in the acknowledgments section of your manuscript. Please also indicate clearly a lead author for this group along with a contact email address.

4. Please note that all PLOS journals ask authors to adhere to our policies for sharing of data and materials: https://journals.plos.org/plosone/s/data-availability. According to PLOS ONE’s Data Availability policy, we require that the minimal dataset underlying results reported in the submission must be made immediately and freely available at the time of publication. As such, please remove any instances of 'unpublished data' or 'data not shown' in your manuscript and replace these with either the relevant data (in the form of additional figures, tables or descriptive text, as appropriate), a citation to where the data can be found, or remove altogether any statements supported by data not presented in the manuscript.

Reviewers' comments:

Reviewer's Responses to Questions

**Comments to the Author**

1. Is the manuscript technically sound, and do the data support the conclusions?

Reviewer #1: Yes

2. Has the statistical analysis been performed appropriately and rigorously? 

Reviewer #1: Yes

3. Have the authors made all data underlying the findings in their manuscript fully available?

Reviewer #1: Yes

4. Is the manuscript presented in an intelligible fashion and written in standard English?

Reviewer #1: Yes

5. Review Comments to the Author

Reviewer #1: The study titled, “Too fast to stay on track? Shorter time to first anti-retroviral regimen is not associated with 2 better retention in care in the French Dat’AIDS cohort” describes factors affecting time from HIV diagnosis to first medical encounter and first ART prescription and influence of an early ART start on 1-year retention in care. The authors report a decrease in time between HIV diagnosis and first medical encounter as well as first ART prescription between 2010 and 2015. However, interestingly, the authors report that initiation of ART within 9 days negatively correlated with retention in care at the end of one year. As the authors have correctly noted, this study has limitations in that patient characteristics, which may relate to retention in care, have not been taken into account.

Minor comments

1. There is only minimal discussion on the possible reasons for early ART initiation relating to lower retention rates. Agreed that qualitative studies are required to identify exact reasons, and the authors have mentioned less time to motivate and prepare the patients as a possible reason. This may be discussed further with additional speculations.

6. PLOS authors have the option to publish the peer review history of their article (what does this mean?). If published, this will include your full peer review and any attached files.

Reviewer #1: No

---

## [Author Response · Author response to Decision Letter 0]

20 Aug 2019

We did our best to answer yours and the reviewer’s queries, and in details:

1. The text was edited to meet PLOS ONE's style requirements

2. A lead author for the “Dat’AIDS Study Group” along with a contact email address has been added and the affiliation of the authors added

3. “Data not shown” has been removed from the text and a supplementary table has been added, along with a brief explicative sentence (line 245)

4. We tried to speculate more on the reasons why very fast ART initiation could be related with poor retention, see additional sentences lines 220-225. As we were truly surprised by this result, we do come short to find an explanation, and are planning to investigate this in depth.

We are willing to make more modifications if necessary.

---

## [Editor Report · Decision Letter 1]

22 Aug 2019

Too fast to stay on track? Shorter time to first anti-retroviral regimen is not associated with better retention in care in the French Dat’AIDS cohort

PONE-D-19-18078R1

Dear Dr. Cuzin,

We are pleased to inform you that your manuscript has been judged scientifically suitable for publication and will be formally accepted for publication once it complies with all outstanding technical requirements.

With kind regards,

Cristian Apetrei, MD, PhD

Academic Editor

PLOS ONE
---

## [Editor Report · Acceptance letter]

28 Aug 2019

PONE-D-19-18078R1 

Too fast to stay on track? Shorter time to first anti-retroviral regimen is not associated with better retention in care in the French Dat’AIDS cohort 

Dear Dr. Cuzin:

I am pleased to inform you that your manuscript has been deemed suitable for publication in PLOS ONE. Congratulations! Your manuscript is now with our production department. 

With kind regards,

on behalf of

Dr. Cristian Apetrei 

Academic Editor

PLOS ONE